# Puerarin Attenuates Obesity-Induced Inflammation and Dyslipidemia by Regulating Macrophages and TNF-Alpha in Obese Mice

**DOI:** 10.3390/biomedicines10010175

**Published:** 2022-01-14

**Authors:** Ji-Won Noh, Hee-Kwon Yang, Min-Soo Jun, Byung-Cheol Lee

**Affiliations:** Department of Clinical Korean Medicine, Graduate School, Kyung Hee University, 26 Kyungheedae-ro, Dongdaemun-gu, Seoul 02447, Korea; oiwon1002@khu.ac.kr (J.-W.N.); yang2019@khu.ac.kr (H.-K.Y.); jms3798@khu.ac.kr (M.-S.J.)

**Keywords:** puerarin, adipose tissue macrophage, inflammation, obesity, molecular docking

## Abstract

Obesity causes low-grade inflammation that results in dyslipidemia and insulin resistance. We evaluated the effect of puerarin on obesity and metabolic complications both in silico and in vivo and investigated the underlying immunological mechanisms. Twenty C57BL/6 mice were divided into four groups: normal chow, control (HFD), HFD + puerarin (PUE) 200 mg/kg, and HFD + atorvastatin (ATO) 10 mg/kg groups. We examined bodyweight, oral glucose tolerance test, serum insulin, oral fat tolerance test, serum lipids, and adipocyte size. We also analyzed the percentage of total, M1, and M2 adipose tissue macrophages (ATMs) and the expression of F4/80, tumor necrosis factor-α (TNF-α), C-C motif chemokine ligand 2 (CCL2), CCL4, CCL5, and C-X-C motif chemokine receptor 4. In silico, we identified the treatment-targeted genes of puerarin and simulated molecular docking with puerarin and TNF, M1, and M2 macrophages based on functionally enriched pathways. Puerarin did not significantly change bodyweight but significantly improved fat pad weight, adipocyte size, fat area in the liver, free fatty acids, triglycerides, total cholesterol, and HDL-cholesterol in vivo. In addition, puerarin significantly decreased the ATM population and TNF-α expression. Therefore, puerarin is a potential anti-obesity treatment based on its anti-inflammatory effects in adipose tissue.

## 1. Introduction

Obesity and dietary excess are associated with hypertrophied adipocytes and high lipolysis, which results in subsequent metabolic disorders such as dyslipidemia, type 2 diabetes, cerebrovascular disease, and cancers. The major link between obesity and its complications is the chronic inflammatory response, which begins with high infiltration of adipose tissue macrophages (ATMs). Adipose tissue inflammation is characterized by a disproportion between pro-and anti-inflammatory immune cell homeostasis, particularly in ATMs [1]. Pro-inflammatory cytokines, including tumor necrosis factor-α (TNF-α) and chemoattractants from adipocytes and ATMs, mediate obesity-induced insulin resistance and liver steatosis. This inflammatory cascade from adipose tissue throughout the body results in obesity-induced complications. Therefore, the aim of new obesity treatment is to reduce weight and improve obesity-induced inflammation.

Puerarin is a major isoflavonoid isolated from *Pueraria lobata (Willd.) Ohwi*, a traditional Chinese medicine used to raise yang qi and replenish the essence of the flesh and muscle. Puerarin is known to have anti-inflammatory, antihypertensive, antidiabetic, antioxidant, anti-apoptotic, cardioprotective, and anticancer effects [2,3,4,5,6]. In in vivo and in vitro studies, puerarin treatment has been shown to lose weight and improve glucose and lipid metabolism in diabetic models [3,4,5]. Regarding the mechanisms underlying the metabolic benefits, puerarin suppressed TNF-α by inhibiting the TLR4/MyD88/NF-κB pathway [6], restored mitochondrial capacity by modulating the PARP-1/PI3K/AKT pathway [5], reduced adipokines, such as leptin and resistin [4], and increased beneficial gut microbiota in obese models [3]. However, few studies have explored the immunomodulatory effects of puerarin on ATM levels. Regarding in silico studies, puerarin was previously researched in Alzheimer’s disease, interaction with cancer macromolecules, and multidrug resistance by autophagy [7,8,9]. No in silico studies have explored the interactions between puerarin and obesity or metabolic dysfunction. Here, we first studied obesity-related target genes of puerarin treatment in silico and investigated the effects of puerarin on obesity and complications, as well as the underlying anti-inflammatory mechanism of ATMs and pro-inflammatory cytokines in vivo.

## 2. Materials and Methods

### 2.1. In Silico Approach for the Anti-Obesity Effects of Puerarin

#### 2.1.1. Puerarin and Its Target Genes

We collected pharmacological data for puerarin using TCMSP (http://old.tcmsp-e.com/, access date: 20 November 2021) and PubChem (https://pubchem.ncbi.nlm.nih.gov/, access date: November 20 2021), and searched the ability to induce or inhibit CYP450 enzymes using ADMET lab (https://admet.scbdd.com/, access date: 20 November 2021). We then identified the target genes of puerarin using SwissTargetPrediction (http://www.swisstargetprediction.ch/, access date: 20 November 2021).

#### 2.1.2. Target Genes Prediction of Obesity

We searched the target genes of obesity using the word “obesity” in three databases (Genecards, OMIM, and DisGeNET). Then, we gathered obesity-targeted genes from Genecards with relevance scores >10 and added the results from the other two databases. We used VENNY 2.1 to determine the overlapping genes in puerarin and obesity using a Venn diagram.

#### 2.1.3. Protein-Protein Interaction (PPI) Network Building

To examine the PPI network for potential obesity-treatment targets of puerarin, we used STRING-DB v11.5, with confidence scores of >0.7 for network construction, and Cytoscape software for network visualization. We extracted 10 genes in the order of high degree from the results of STRING.

#### 2.1.4. Gene Ontology (GO) and Target Pathway

We gathered the results of functional enrichment analysis in GO and the Kyoto Encyclopedia of Genes and Genomes (KEGG) pathway from the Database for Annotation, Visualization, and Integrated Discovery (DAVID) v6.8 (access date: 20 November 2021) and visualized the data using R software as dot plots.

#### 2.1.5. Docking Simulation

To prepare the molecular structure of the ligand, we obtained the 3D structure of puerarin (CID: 5281807) from PubChem and converted it to the PDB and PDBQT format using PyMOL and AutoDock, respectively. Next, we collected the molecular structures of three selected receptors: TNF-α (PDB ID: 2AZ5), M1 macrophages (PDB ID: 1GD0), and M2 macrophages (PDB ID: 1JIZ) from the Protein Data Bank (https://www.rcsb.org, access date: 20 November 2021). We utilized AutoDock Vina to simulate ligand-receptor docking and calculate the binding energy, and the Biovia Discovery Studio Visualizer was used to visualize the docking formation.

### 2.2. In Vivo Approach to Evaluate the Anti-Obesity Effects and Metabolic Benefits of Puerarin

#### 2.2.1. Preparation of Animal Models and Study Design

The male C57BL/6 mice (Central Lab Animal, Seoul, Korea) used in the experiment were 6-weeks-old and weighed 19–22 g. The mice had access to food and water ad libitum, and their environment was controlled to maintain light and dark cycles every 12 h at 40–70% humidity. Twenty mice were divided into four groups: normal chow (NC), control, puerarin 200 mg/kg (PUE), and atorvastatin 10 mg/kg (ATO) groups. Bodyweight (BW) was monitored weekly before the morning feeding. To induce obesity, except for the NC group, a 60% fat diet was provided for 8 weeks. At 8 weeks, when a significant weight gain was confirmed in the control group (*p* < 0.001), the mice in the PUE, ATO, and control groups were orally administered puerarin 200 mg/kg, atorvastatin 10 mg/kg, or normal saline once daily. The puerarin used in this study was purchased from Tokyo Chemical, Inc. Ltd., Tokyo, Japan) and had a purity higher than 96%, as measured by nuclear magnetic resonance, high-performance liquid chromatography, and mass spectrometry. Atorvastatin was purchased from Sigma-Aldrich (St. Louis, MO, USA). The Kyung Hee Medical Animal Research Ethics Committee (KHMC-IACUC 2020-014, approval date: 12 April 2021) approved all procedures for animal experiments.

#### 2.2.2. Assessment of Lipid Metabolism and Safety Profile

We performed an oral fat tolerance test (OFTT) 14 weeks after 14 h of fasting. The first TG measurement from the tail vein blood was conducted immediately after fasting, and 2 g/kg of olive oil (Sigma, St. Louis, MO, USA) was orally administered. Blood samples were collected from the tail veins at 120-, 240-, and 360-min following TG measurements. We used Accutrend Plus (Roche, Basel, Switzerland) and AccuTrend triglyceride strips using a triglyceride colorimetric assay kit (Cayman, Ann Arbor, MI, USA) to measure TG levels in OFTT. 

Before the mice were sacrificed with ethanol anesthesia, we obtained blood samples from the heart to examine serum lipids and liver and kidney functions, including non-esterified fatty acids (NEFA), triglycerides (TG), total cholesterol (TC), low-density lipoprotein cholesterol (LDL-C), high-density lipoprotein cholesterol (HDL-C), phospholipid levels, aspartate aminotransferase (AST), alanine aminotransferase (ALT), and creatinine. We used an ELISA kit (MyBioSource, San Diego, CA, USA) to analyze biochemical parameters.

#### 2.2.3. Assessment of Glucose Metabolism and Insulin Resistance 

At 13 weeks, fasting blood glucose levels were examined after 14 h of fasting. In the oral glucose tolerance test (OGTT), mice were orally fed glucose (2 g/kg bodyweight) diluted in distilled water, and blood glucose levels were measured at 30-, 60-, 120-, and 180-min. Blood glucose was measured using a strip-operated blood glucose sensor (ACCU-CHECK Performa, Roche, Basel, Switzerland).

At 16 weeks, insulin measurement was performed using blood samples obtained after 6 h of fasting using an ultrasensitive mouse insulin ELISA kit (Crystal Chem Inc., IL, USA). We used BD Microtainer serum separator tubes for blood collection and 96-well antibody-coated microplates (5 μL each) for plating the samples and insulin standards. After 2 h of incubation and washing, the anti-insulin enzyme conjugate and enzyme-substrate solution were added to each well and incubated [10]. Ten minutes after the reaction stop solution was added, the absorbance was read at 450 nm, and homeostatic model assessment for insulin resistance (HOMA-IR) value was obtained using the following equation: HOMA-IR = fasting blood glucose (mg/dL) × fasting insulin (ng/mL) × 0.071722516669606. 

#### 2.2.4. Histologic Analysis 

The liver and adipose tissue samples were fixed in 10% neutral-buffered formalin. The samples were dehydrated in 70%, 80%, 95%, and 100% ethanol, embedded in paraffin, and rested on gelatin-coated 4-μm thick slides. The slices were dewaxed in xylene, rehydrated in 100%, 95%, 80%, and 70% distilled water, and stained with hematoxylin and eosin (H & E). A high-resolution camera-mounted optical microscope (Olympus BX-50, Olympus Optical, Nagano, Japan) was used to capture digital images. We calculated the adipocyte size and fat area in the liver using the ImageJ software [10].

#### 2.2.5. Extraction of RNA and Real-Time PCR 

Epididymal fat samples were packaged with aluminum foil and stored in liquid nitrogen at −70 °C. RNA was extracted from the samples using Mini RNA Isolation IITM (ZYMO Research, Irvine, CA, USA), crushed in 300 μL ZR RNA buffer with a homogenizer, and centrifuged. Then, we collected the supernatant, centrifuged the 2 mL Zymo-Spin Ш columns at 2000 rpm, washed them twice with 350 μL of wash buffer, and centrifuged for 1 min. The samples containing RNase-free water in 1.5 mL collection tubes were centrifuged at 1000 rpm and stored at −70 °C [10]. 

Quantitative real-time polymerase chain reaction (qRT-PCR) was used to quantify inflammatory gene expression in adipose tissue, including TNF-α, F4/80, C-C motif ligand 2 (CCL2), C-C motif ligand 4 (CCL4), CCL5 (RANTES), and C-X-C motif chemokine receptor 4 (CXCR4). An Advantage RT PCR Kit (Clontech, Palo Alto, CA, USA) was used for complementary DNA (cDNA) synthesis. RNA (1 μg) was cultured at 70 °C for 2 min in a mixed state with oligo (dT) and RNase-free H_2_O, and then cultured at 42 °C for 60 min and at 94 °C for 5 min after mixing with MMLV reverse transcriptase, 5× reaction buffer, recombinant RNase inhibitor, and 10 nM dNTP. We performed reverse transcription PCR to obtain cDNA: dH_2_O, primers, and 2× SYBR Reaction buffer and used a 7900HT Fast Real-Time PCR System (Applied Biosystems, Waltham, MA, USA) for qRT-PCR. The primer sequences were as follows: TNF-α, 5′-TTCTG TCTAC TGAAC TTCGG GGTGA TCGGT CC-3′, and 5′-484 GTATG AGATA GCAAA TCGGC TGACG GTGTGGG-3′; F4/80, 5′-CTTTGGC-485 TATGGGCTTCCAGTC-3′ and 5′-GCAAGGAGGACAGAGTTTATCGTG-3′; CCL2, 5′-486 AGGTCCCTGTCATGCTTCTGG-3′, and 5′-CTGCTGCTGGTGATCCTCTTG-3′; CCL4, 5′-487 CTCAGCCCTGATGCTTCTCAC-3′, and 5′-AGAGGGGCAGGAAATCTGAAC-3′; CCL5, 488 5′-TGCCCACGTCAAGGAGTATTTC-3′ and 5′- AACCCACTTCTTCTCTGGGTTG-3′; 489 CXCR4, 5′-TCAGTGGCTGACCTCCTCTT-3′ and 5′-CTTGGCCTTTGACTGTTGGT-3′; 490 and glyceraldehyde-3-phosphate dehydrogenase (GAPDH, housekeeping gene), 5′-AG-491 TCCATGCCATCACTGCCACC-3′ and 5′- CCAGTGAGCTTCCCGTTCAGC-3′. For gene expression analysis, we converted the cycle threshold (Ct) calculated by SDS Software 2.4 (Applied Biosystems^®^, USA) to relative quantitation (RQ) based on GAPDH. The fold-change value was adjusted by considering the NC group value as 1.

#### 2.2.6. Isolation of Stromal Vascular Cells (SVCs) and Fluorescence-Activated Cell Sorting (FACS) Evaluation of ATMs 

We cut the fat tissue samples in a mixed state with phosphate-buffered saline (PBS; Gibco, MA, USA) and 2% bovine serum albumin (BSA; Gibco, Grand Island, NY, USA) into 1–2 mm pieces. The samples were shaken at 37 °C for 20–25 min after adding collagenase (Sigma-Aldrich, St. Louis, MO, USA) and DNase I (Roche, USA). Further, we added 5 nM EDTA, filtered the samples through a 100 μm filter (BD Biosciences, San Diego, CA, USA), and centrifuged at 1000 rpm for 3 min. The pellet that remained after removing the upper solution was mixed with PBS and 2% fetal bovine serum (FBS; Sigma, St. Louis, MO, USA) and centrifuged at 200× *g* for 10 min. SVCs were obtained from the bottom of the tubes. Each sample was modulated to obtain 106 cells using a Cellometer (Nexcelom Bioscience LLC, Lawrence, MA, USA) and rested for 10 min to react with the added FcBlock (BD Pharmingen, San Diego, CA, USA) at a ratio of 1:100. The samples were then rested with fluorophore-conjugated antibodies in the dark for 20 min [10]. We used the following antibodies: CD45-APC Cy7 (BioLegend, San Diego, CA, USA), F4/80-APC (BioLegend, San Diego, CA, USA), CD11c-phycoerythrin (CD11c-PE, BioLegend, USA), and CD206-FITC (BioLegend, San Diego, CA, USA). The samples were washed with 2% FBS/PBS solution, centrifuged at 1500 rpm, and transferred to fluorescence-activated cell sorting (FACS) tubes. We used a FACS Calibur (BD Biosciences, San Diego, CA, USA) and estimated the percentage of CD45(+) F4/80(+), CD45(+) F4/80(+) CD11c(+), and CD45(+) F4/80(+) CD206(+) macrophages using FlowJo (Tree Star, Inc., Ashland, OR, USA).

#### 2.2.7. Statistical Analysis

Statistical analysis of experimental data was performed using GraphPad PRISM 5 (GraphPad Software Inc., San Diego, CA, USA) and represented as the mean ± standard error of the mean (SEM). A one-way analysis of variance and Tukey’s post-hoc test were used to analyze the statistical significance of the between-group differences. Asterisks (*) indicate significant differences compared to the NC group and number signs (#) indicate the difference compared to the control group; *, # for *p* < 0.05; **, ## for *p* < 0.01; and ***, ### for *p* < 0.001.

## 3. Results

### 3.1. In Silico Analysis of Anti-Obesity Effects of Puerarin

#### 3.1.1. Ingredient Target Genes and PPI Network

Pharmacokinetic analysis showed that puerarin had 24.03% oral bioavailability, 0.69% drug-likeness, and −1.15 of Caco-2, and, in terms of metabolism, did not inhibit CYP450 1A2, 3A4, 2C9, 2C19, and 2D6. We identified 103 target genes of puerarin and 2915 obesity-related genes. There were 69 overlapping genes (Figure 1a,b). In the PPI analysis, the top 10 genes with high connectivity were selected (Figure 1c).

#### 3.1.2. GO and KEGG Pathway Enrichment Analysis

GO showed 59 molecular function (MF) terms, 211 biological process (BP) terms, and 30 cellular component (CC) terms. The top 7 and 10 enriched terms were selected for each category based on gene counts >10. We constructed a dot plot with 24 terms (Figure 1d). The MF category included protein binding, protein kinase binding, ATP binding, and enzyme binding. The BP category included oxidation-reduction process, inflammatory response, and positive regulation of the ERK1 and ERK2 cascade. Based on gene count, we identified the top 29 pathways in the KEGG pathway enrichment analysis (Figure 1e), which were associated with insulin resistance and inflammation, such as TNF signaling, PI3K-Akt signaling, MAPK signaling, HIF-1 signaling, estrogen signaling, cAMP signaling, and mTOR signaling pathways. 

#### 3.1.3. Docking

The results from the GO and KEGG pathway analyses were commonly involved in the immune response in adipose tissue, especially macrophages. Therefore, we checked and confirmed high-affinity binding between puerarin and TNF (PDB ID: 2AZ5), M1 macrophages (PDB ID: 1GD0), and M2 macrophages (PDB ID: 1JIZ), and visualized molecular docking interactions using Biovia Discovery Studio Visualizer (Figure 2). The binding energy of each interaction was high at −7.6, −8.4, and −9.7 kcal/mol.

### 3.2. In Vivo Examination of Anti-Obesity Effects of Puerarin 

#### 3.2.1. Body, Liver, and Fat Weights

Significant increases in bodyweight and liver and epididymal fat pad weights were noted in the control group compared to the NC group. This effect on epididymal fat weights was significantly attenuated in the PUE group (1.99 ± 0.16 vs. 1.40 ± 0.03, *p* < 0.01) (Figure 3c). However, PUE administration showed no significant difference in body and liver weights compared to the control group (Figure 3a,b). There was no significant difference in food intake between the groups fed a high-fat diet. 

#### 3.2.2. Serum Lipids and Lipid Metabolism

Significant increases in serum lipids, including NEFA, TC, LDL-C, TG, and phospholipids, were observed in the control group compared to the NC group. The PUE group significantly reduced the NEFA, TC, and TG levels compared to the control group (NEFA, 2190.2 ± 24.99 mEq/L vs. 2521.8 ± 1.5 mEq/L, *p* < 0.001; TC, 179.8 ± 9.54 mg/dl vs. 234.4 ± 19.38 mg/dl, *p* < 0.05; TG 131.8 ± 4.95 mg/dl vs. 162.2 ± 8.13 mg/dl, *p* < 0.05) (Figure 4a,b,e). However, LDL-C levels did not significantly decrease in the PUE group (Figure 4c). The PUE group showed significantly increased HDL-C and phospholipid levels, OFTT, and AUC compared to the control group (Figure 4d,f,g). 

#### 3.2.3. Insulin Resistance and Glucose Metabolism

In the OGTT results, the glucose levels at all time points and AUC were significantly higher in the control group than in the NC group (AUC: 45741 ± 800.03 vs. 32127 ± 1073.23, *p* < 0.001). However, the PUE group displayed no significant differences compared to the control group (Figure 4h,i). All experimental groups showed no significant differences in FBG levels (Figure 4j). As obesity was induced in the control group, insulin and HOMA-IR were significantly elevated, but the PUE group showed no significant advantage in both values (Figure 4k,l).

#### 3.2.4. Adipose Tissue Macrophages

The control group displayed significantly increased total ATMs and decreased CD206+ ATMs compared to the NC group. Total ATMs were significantly reduced in the PUE group compared to the control group (54.03 ± 0.93 vs. 79.84 ± 2.89, *p* < 0.001) (Figure 5a,b), but CD206+ ATMs exhibited no significant difference between HFD-fed groups (Figure 5d). The CD11c+ ATM population increased in the control group compared to the NC group and decreased in the PUE and ATO groups compared to the control group (Figure 5c). 

#### 3.2.5. Inflammatory Cytokines 

The control group displayed a significant increase in the expression of all inflammatory genes, including TNF-α, F4/80, CCL2, CCL4, CCL5 (RANTES), and CXCR4, compared to the NC group. PUE significantly reduced the expression of F4/80 and TNF-α (F4/80: 4.24 ± 0.71 vs. 8.27 ± 1.37, *p* < 0.05; TNF-α: 6.59 ± 0.70 vs. 10.28 ± 1.13, *p* < 0.05) (Figure 6). The PUE group showed lower CCL2, CCL4, CCL5, and CXCR4 expression than the control group (Figure 6). ATO significantly reduced CCL2 expression. 

#### 3.2.6. Size of Adipocytes and Fat Area in the Liver

The adipocytes and fat area in the liver in the control group was significantly larger than that in the NC group. We observed that the PUE group showed significantly down-sized adipocytes both in the liver and fat pads compared to the control group (fat: 1625 ± 105.1 vs. 2713 ± 177.4, *p* < 0.001; liver: 25.29 ± 1.016 vs. 96.98 ± 5.907, *p* < 0.001) (Figure 7). Adipocyte size in the ATO group was significantly decreased in the liver and increased in fat pads.

#### 3.2.7. Safety of Puerarin Administration

AST and ALT levels were elevated in the control group compared to the NC group, and those in the PUE group were not different from those in the control group (Figure 3d,e). The serum creatinine level was significantly decreased only in the ATO group compared to the control group (Figure 3f).

## 4. Discussion

This study is the first to investigate the anti-obesity effect of puerarin against adipose tissue inflammation, especially in the context of macrophages. These findings suggest that puerarin treatment attenuates ATM recruitment, reduces differentiation towards M1 ATMs, and suppresses pro-inflammatory cytokines. In addition, puerarin improved the serum lipid profile, specifically NEFA, TC, HDL-C, TG, and phospholipids. It decreased the size of the adipocytes and the fat area in the liver without any change in bodyweight. Taken together, puerarin improved lipid metabolism in HFD-induced obese mice by modulating macrophages and pro-inflammatory cytokines in adipose tissue. In addition, puerarin is reasonable to consider as a combination therapy with conventional anti-diabetic drugs since it did not inhibit any of CYP450 enzymes. 

The in silico approach to explore the anti-obesity effects of puerarin suggested that the top 10 hub genes, including TNF, EGFR, MMP9, AKT1, PIK3CA, HIF1A, PRKCZ, TP53, ESR1, and ETGS2, may be the treatment target genes with anti-obesity effects. MMP9 expression in adipose tissue is positively related to HOMA-IR and reduces diet-induced insulin resistance [11]. High free fatty acids activate epidermal growth factor receptor (EGFR). EGFR signaling activates NF-κB signaling, which is critical for pro-inflammatory cytokine production by macrophages [12], and EGFR knockout mice showed suppression of M1 polarization and Th17 responses [13]. PTGS2 (Prostaglandin-endoperoxide synthase 2) is expressed in M1 macrophages and promoted by elevated glucose levels. In ATMs, inhibition of PTGS2 suppresses M1 polarization and the production of TNF-α and CCL2 [14]. PIK3CA provides indications for the p110a catalytic subunit of PI3K (phosphatidylinositol 3-kinase) [15]. Loss of p110a in adipose tissue increases adiposity, glucose intolerance, and liver steatosis by reducing energy expenditure [16]. The PI3K-Akt pathway modulates macrophage activation and is associated with an anti-inflammatory macrophage response [17]. AKT1 is known to promote M2 differentiation via the PI3K/AKT/mTOR signaling cascade [18]. TP53 expression stimulates macrophage polarization in the M2-state [19]. HIF1A (Hypoxia-inducible factor 1α) is responsible for inflammatory polarization of macrophages in a hypoxic state [20]. Estrogen receptor alpha (ESR1) regulates angiogenesis of adipose tissue via VEGFA [21], and ESR1-deficient mice are exposed to insulin resistance and obesity [22]. Taken together, we found that the obesity-related target genes of puerarin were commonly associated with macrophages in inflammation and metabolic dysfunction in glucose and lipids. This finding was the basis of our molecular docking with puerarin and TNF and two subtypes of macrophages. We discovered a high binding affinity between puerarin and TNF, M1, and M2 macrophages. Therefore, we established the following in vivo experiments and examined the gene expression of TNF and several chemokines related to macrophage recruitment and activation.

Obesity leads to adipocyte hypertrophy and increases pro-inflammatory cytokine levels in the adipose tissue. Chronic adipose tissue inflammation begins with high infiltration of ATMs, which leads to metabolic complications, such as dyslipidemia, non-alcoholic fatty liver disease (NAFLD), and insulin resistance [23]. ATMs, characterized by CD45+ F4/80+, are recruited to adipose tissue in response to CCL2 from hypertrophied adipocytes [24]. In our study, puerarin treatment significantly reduced the total ATM population, suggesting inflammatory attenuation. ATMs have two phenotypes: classically activated M1 manifests as F4/80+ CD11c+, while the alternatively activated M2 manifests as F4/80+ CD206+. However, as there are several definitions of macrophage activation, the mechanical division of M1 and M2 has not reached an agreement and many scientists have recently argued about the inconsistent use of macrophage surface markers [25]. In the obese state, ATM polarization is shifted towards M1 and M1 ATMs, which mainly upregulates monocytes’ chemoattractants, enhances adipocyte lipolysis and induces insulin resistance by secreting TNF-α and IL-6 [1,26,27]. We found that oral administration of puerarin decreased the proportion of M1 ATMs and significantly suppressed TNF-α expression, suggesting inflammatory attenuation. Other pro-inflammatory chemokines that are highly expressed in obese adipose tissue recruit immune cells and activate them to secrete pro-inflammatory cytokines [28]. CCL2, also called MCP-1, plays a powerful role in ATM recruitment [29,30]. CCL4, also called MIP-1b, and CCL5 (RANTES) are highly expressed in obese ATMs and are powerful chemokines that attract monocytes, memory T cells, and natural killer cells [31]. CCL5 (RANTES) has recently been shown to regulate M2 polarization [28,30]. CCL2 and CCL5 have been reported to increase, particularly in the ATM fraction, with obesity, and are correlated with T cells in adipose tissue. CXCR4 induces ATM infiltration and M1 macrophage differentiation; thus, suppressing this gene reduces pro-inflammatory cytokines [29,30]. Here, puerarin significantly suppressed the expression of F4/80 and TNF-α, suggesting attenuation of adipose tissue inflammation. The PUE group also showed decreased expression of CCL2, CCL4, CCL5, and CXCR4, although the results were not significant.

Puerarin is known to have therapeutic properties in obesity [32] and its related metabolic complications [33]. Regarding the underlying mechanisms, puerarin has been reported to alleviate inflammatory responses in diabetes [6], non-alcoholic liver disease [33,34], and atherosclerosis [2,35]. However, the anti-inflammatory mechanisms in adipose tissue are not fully understood. At the macrophage level, few studies have explored the effect of puerarin on ATMs. Although it was not an obesity model, Peng et al. [34] reported the inhibitory effect of puerarin on Kupffer cells in the alcoholic liver. We first investigated the immunomodulation of ATMs by puerarin and observed a significant reduction in total ATMs. In addition, the PUE group had decreased CD11c+ M1 ATM levels, but this result was not significant. Our findings are consistent with those of an in vitro study demonstrating the TNF-α inhibitory effect of puerarin in RAW264.7 cells by downregulating P2X4R-mediated inflammatory signaling [36]. Increased free fatty acids and TNF-α from adipocytes and ATMs activate NF-κB and JNK signaling pathways, mediating obesity-induced insulin resistance and lipid accumulation in the liver [37]. Wenting et al. [6] observed that puerarin treatment reduced TNF-α expression by inhibiting the TLR4/MyD88/NF-κB pathway in adipose tissue, and Shuai et al. [33] reported that puerarin reduced hepatic fat accumulation by attenuating PARP-1 in NAFLD. In an in vitro study, Xiao et al. [38] observed suppression of TNF-α and NF-κB expression in peripheral blood mononuclear cells after puerarin administration. In this study, puerarin reduced TNF-α and NEFA levels and decreased the fat area in the liver. Taken together, puerarin treatment suppressed adipose tissue inflammation at the macrophage level by modulating polarization and inflammatory cytokine secretion via the TNF-α/NF-κB pathway [39].

The lipid-lowering effect on TC, TG, and LDL-C and the weight loss effect of puerarin have been reported in previous in vivo studies [4,40,41]. In this study, puerarin treatment significantly reduced serum NEFA, TC, TG, and phospholipid levels, increased HDL-C levels, and improved all time points of OFTT. However, the PUE group showed a significant weight reduction in epididymal fat pads without bodyweight loss. Taken together, the effects of puerarin on lipid metabolism were based on an anti-inflammatory mechanism, specifically in ATMs, and not on weight loss. Furthermore, this study established a positive control group with oral administration of atorvastatin, an HMG-CoA reductase inhibitor. The ATO group also improved the serum lipid profile but had no effect on adipocyte size and TNF-α expression. Unlike the ATO group, the PUE group showed significantly smaller adipocytes and lower expression of F4/80 and TNF-α compared to the control group. Compared to atorvastatin, the lipid-lowering effect of which is based on the mechanism of activation of LDL receptors in the liver, puerarin alleviates adipose tissue inflammation.

Regarding the effect on insulin resistance, previous studies reported that puerarin and Radix pueraria lobata improved glucose metabolism by reducing FBG and insulin concentrations. However, we observed no significant improvement in FBG, HOMA-IR, and OGTT results. Obesity-induced insulin resistance is indirectly regulated by an imbalance in lipid homeostasis in the adipose tissue [1]. We established a short treatment period of 8 weeks but showed improved lipid metabolism and adipose tissue inflammation. In future studies, it should be confirmed that longer puerarin treatment improves glucose metabolism.

## 5. Conclusions

This study is the first to observe the anti-inflammatory effects of puerarin on ATMs. We showed that oral administration of puerarin in HFD-induced obese mice reduced fat accumulation in adipose tissue and liver without bodyweight loss, improved lipid metabolism, and decreased total ATMs, M1 ATMs, and pro-inflammatory cytokines. Therefore, puerarin has the potential to treat obesity and its related complications by modulating ATM-induced inflammation by suppressing the TNF-α/NF-κB pathway.

## Figures and Tables

**Figure 1 biomedicines-10-00175-f001:**
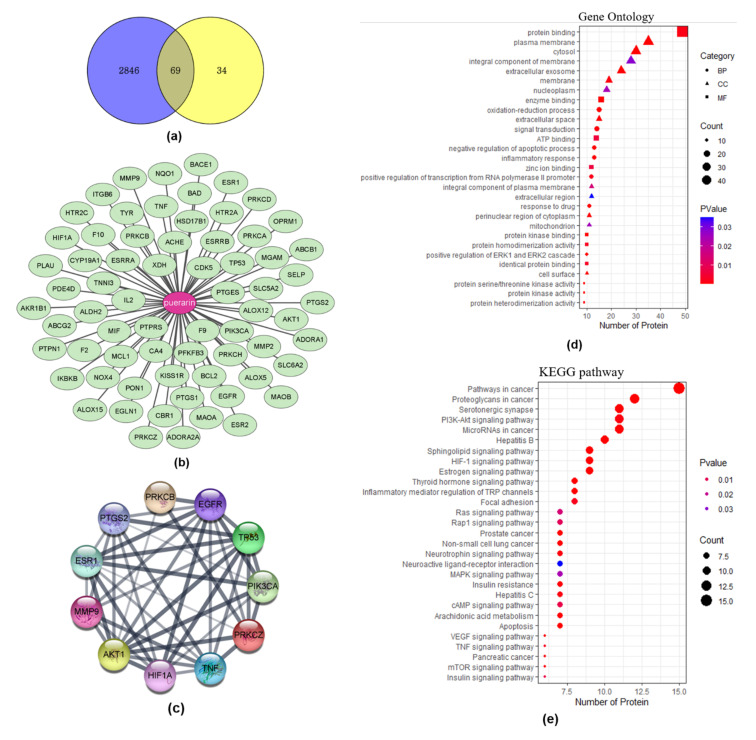
PPI network between hub genes of puerarin and obesity and functional pathway analysis. (**a**) Sixty-nine overlapping genes. (**b**) Ligand-target protein network. (**c**) Top 10 genes at high levels of connectivity. (**d**) GO enrichment analysis. (**e**) KEGG pathway analysis. PPI: Protein-protein interaction, GO: Gene ontology, KEGG: Kyoto Encyclopedia of Genes and Genomes.

**Figure 2 biomedicines-10-00175-f002:**
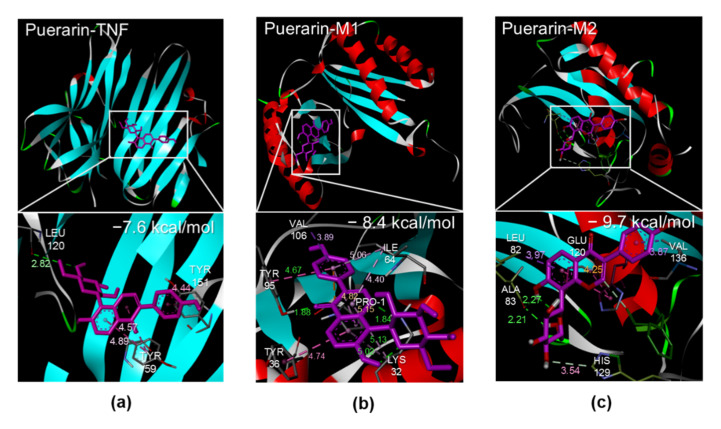
Molecular docking. (**a**) Puerarin and TNF. (**b**) Puerarin and M1 macrophage. (**c**) Puerarin and M2 macrophage. TNF (PDB ID: 2AZ5), M1 macrophage (PDB ID: 1GD0), and M2 macrophage (PDB ID: 1JIZ). The binding energy is written at the top right.

**Figure 3 biomedicines-10-00175-f003:**
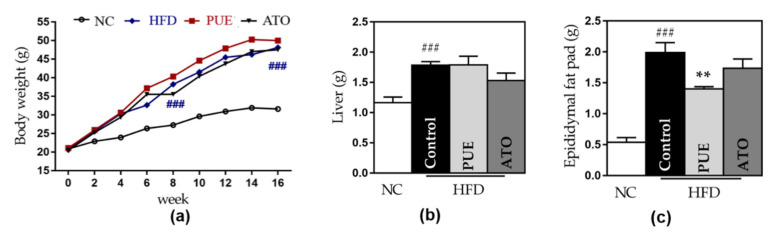
Weight- and safety-related outcomes. (**a**) Bodyweight. (**b**) Liver weight. (**c**) Epididymal fat pads weight and levels of (**d**) AST, (**e**) ALT, and (**f**) creatinine. Data represent mean ± standard error of the mean (SEM). # *p* < 0.05, ## *p* < 0.01 and ### *p* < 0.001 versus the NC group, and * *p* < 0.05 and ** *p* < 0.01 versus the control group. AST: aspartate aminotransferase, ALT: alanine aminotransferase, NC: normal chow, HFD: high fat diet, PUE: puerarin, ATO: atorvastatin.

**Figure 4 biomedicines-10-00175-f004:**
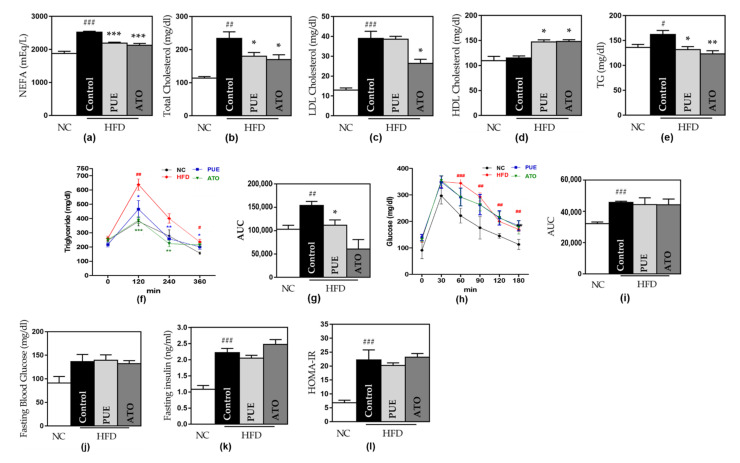
Lipid and glucose metabolism. (**a**) NEFA, (**b**) total, (**c**) LDL cholesterol, (**d**) HDL cholesterol, (**e**) TG, (f) OFTT, (**g**) AUC of OFTT, (**h**) OGTT, (**i**) AUC of OGTT, (**j**) fasting blood glucose, (**k**) fasting insulin, and (**l**) HOMA-IR. Data represent mean ± standard error of the mean (SEM). # *p* < 0.05, ## *p* < 0.01 and ### *p* < 0.001 versus the NC group, and * *p* < 0.05, ** *p* < 0.01 and *** *p* < 0.001 versus the control group. NEFA: non-esterified fatty acid, LDL: low-density lipoprotein, HDL: high-density lipoprotein, TG: triglyceride, AUC: area under the curve, HOMA-IR: homeostatic model assessment for insulin resistance, NC: normal chow, HFD: high fat diet, PUE: puerarin, ATO: atorvastatin.

**Figure 5 biomedicines-10-00175-f005:**
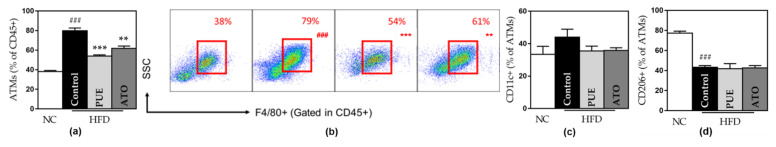
ATM populations. (**a**) CD45 + ATMs. (**b**) Flow cytometry result of CD45 + ATMs. (**c**) CD11c + ATMs, and (**d**) CD206 + ATMs. Data represent mean ± standard error of the mean (SEM). ### *p* < 0.001 versus the NC group, and ** *p* < 0.01 and *** *p* < 0.001 versus the control group. ATMs: adipose tissue macrophages, NC: normal chow, HFD: high rat diet, PUE: puerarin, ATO: atorvastatin.

**Figure 6 biomedicines-10-00175-f006:**
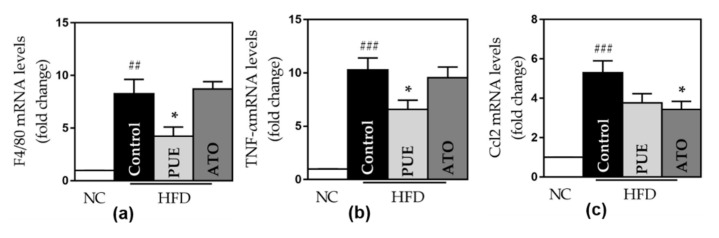
Pro-inflammatory gene expression. The expression of (**a**) F4/80, (**b**) TNF- α, (**c**) CCL2, (**d**) CCL4, (**e**) CCL5, and (**f**) CXCR4. Data represent mean ± standard error of the mean (SEM). # *p* < 0.05, ## *p* < 0.01 and ### *p* < 0.001 versus the NC group, and * *p* < 0.05 versus the control group. ATMs: adipose tissue macrophages, NC: normal chow, HFD: high rat diet, PUE: puerarin, ATO: atorvastatin.

**Figure 7 biomedicines-10-00175-f007:**
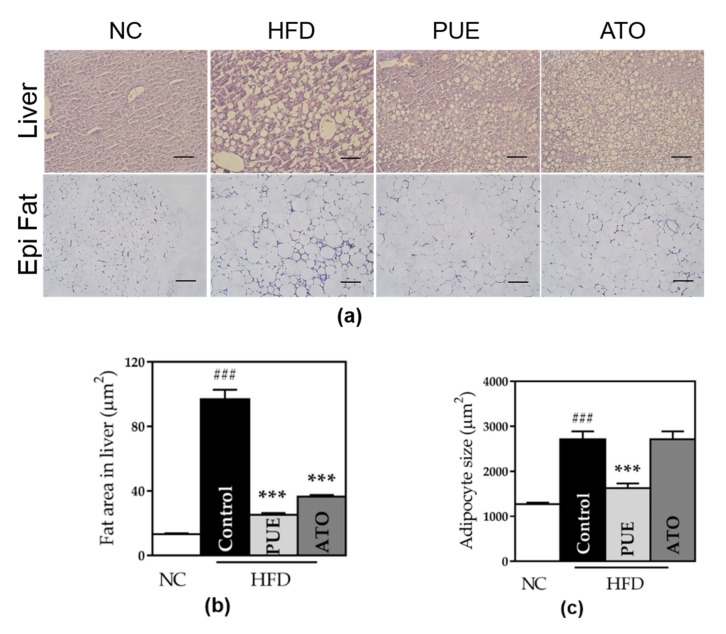
Analysis of tissue microscopic results. (**a**) Histological images of liver and epididymal fat. (**b**) Fat area in the liver and (**c**) adipocyte size. Representative histological images were stained by hematoxylin and eosin (H & E) and the scale bar indicates 5 μm in liver and 100 μm in epi fat. Data represent mean ± standard error of the mean (SEM). ### *p* < 0.001 versus the NC group and *** *p* < 0.001 versus the control group. NC: normal chow, HFD: high fat diet, PUE: puerarin, ATO: atorvastatin.

## Data Availability

Not applicable.

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
