# Peer review of "Puerarin Attenuates Obesity-Induced Inflammation and Dyslipidemia by Regulating Macrophages and TNF-Alpha in Obese Mice"

_biomedicines, 2022, doi:10.3390/biomedicines10010175_

Round 1

Reviewer 1 Report

The manuscript addresses a very important issue of low-grade inflammation resulting in dyslipidemia and insulin resistance. The authors studied the effect of puerarin both in silico and in vivo in mouse model system. They collected the data on body weight, oral glucose tolerance, the level of insulin, lipids, and adipocyte size. They also analyzed the M1, and M2 adipose tissue macrophage expression of F4/80, TNF-α, CCL2, CCL4, CCL5, and C-X-C motif chemo-18 kine receptor 4. They also identified In silico, the puerarin-targeted genes. they found that puerarin improved fat pad weight, adipocyte size, fat area in the liver, free fatty acids, triglycerides, total cholesterol, and HDL-22 cholesterol. Also, the puerarin decreased the expression of TNFand ATM population. The conclusion was that puerarin has an anti-inflammatory effect and therapeutic potential for the treatment of obesity. There are important findings that should be published. The paper is very well written (requires minor editorial work) and has beautiful illustrations.

Author Response

Dear Editor:

 Thank you very much for considering our manuscript for publication. The comments from the reviewers were very helpful to us. In submitting the revised manuscript, we have incorporated their suggestions and comments. We are very pleased to resubmit the revised manuscript entitled “Puerarin attenuates obesity-induced inflammation and dyslipidemia by regulating macrophages and TNF-alpha in obese mice” by Ji-Won Noh et al.

  1. The comments of each reviewer were incorporated, and adequate explanations regarding the results were made in response to the suggestions and questions of each reviewer.
  2. As much of the original manuscript as possible was maintained during the revision.

  We thank you in advance for your consideration of this manuscript.

                                               Yours faithfully,

Byung-Cheol Lee, M.D., Ph.D.

Reviewer 2 Report

This study examines the effect of puerarin on obesity-related inflammation. A number of modern methods were used in the work. Despite the obvious advantages, a number of questions and comments arose.

  1. For statistical analysis, the authors used one-way analysis of variance, which assumes a normal distribution of features. However, such indicators as the relative expression of mRNA, especially normalized to the level of relative expression in the control group, cannot be considered as traits, the distribution of which corresponds to the law of normal distribution. In this connection, the statistical analysis must be carried out using other criteria.
  2.  The authors need to clarify against which marker they used the CD11b or CD11c antibodies. On page 4, line 188, different antibodies are indicated. 
  3. The authors provide microphotographs of histological sections of the liver and adipose tissue. It is necessary to bring better quality photos with higher resolution. 
  4. In connection with the use of markers M1 and M2 of the macrophage phenotype. Indeed, CD206 is considered as a marker of M2 macrophages, but CD11b and even more so CD11c cannot be considered as a marker of M1. In this regard, the following question also arises. What kind of cell population did the authors with the F4/80+CD11c+ phenotype study? Could it be dendritic cells? 
  5. Another issue concerns discussion. The authors point out that infiltrating macrophages have a pro-inflammatory phenotype. How did the authors show that these are precisely the infiltrating macrophages of adipose tissue? Is there a difference in the phenotype of resident macrophages and those migrating in adipose tissue? I think that on the basis of the available markers, the authors cannot reliably speak about infiltrating macrophages, just as they cannot reliably speak about a change in the M1 or M2 phenotype of adipose tissue macrophages. 

Author Response

Thank you very much for considering our manuscript for publication. Your suggestions were very helpful to us, and we have incorporated those points into our revised manuscript.

The changes made to the manuscript are as follows:

[Reviewer 2]

This study examines the effect of puerarin on obesity-related inflammation. A number of modern methods were used in the work. Despite the obvious advantages, a number of questions and comments arose.

  1. For statistical analysis, the authors used one-way analysis of variance, which assumes a normal distribution of features. However, such indicators as the relative expression of mRNA, especially normalized to the level of relative expression in the control group, cannot be considered as traits, the distribution of which corresponds to the law of normal distribution. In this connection, the statistical analysis must be carried out using other criteria.
  • As you commented, we carried out statistical analysis again using ANOVA or Mann-Whitney test depending on the normal distribution or not, and the results of both analyses were not different from each other. So, we added the Mann-Whitney test in the statistical analysis part in the methods section.

  1.  The authors need to clarify against which marker they used the CD11b or CD11c antibodies. On page 4, line 188, different antibodies are indicated. 
  • We corrected CD11b to CD11c. It was a typo that different antibodies were written.

  1. The authors provide microphotographs of histological sections of the liver and adipose tissue. It is necessary to bring better quality photos with higher resolution. 
  • We changed the microphotographs of histological sections ‘Figure 6a’ with higher resolution.

  1. In connection with the use of markers M1 and M2 of the macrophage phenotype. Indeed, CD206 is considered as a marker of M2 macrophages, but CD11b and even more so CD11c cannot be considered as a marker of M1. In this regard, the following question also arises. What kind of cell population did the authors with the F4/80+CD11c+ phenotype study? Could it be dendritic cells? 
  • We used F4/80+CD11c+ cells as M1 macrophages, which is different from dendritic cells (DCs) marked as F4/80-CD11c+. As Lee (2014) reviewed, CD11c is classically considered as a marker of DCs, but, in obese adipose tissue, the CD11c+ adipose tissue macrophages (ATMs) also express F4/80 distinct from adipose tissue DCs. Thus, CD11c has been used as an M1 ATM marker. The following references are other previous studies analyzing M1 marker as CD11c positive.

Related References

- Byung-Cheol Lee, Jongsoon Lee. Cellular and molecular players in adipose tissue inflammation in the development of obesity-induced insulin resistance. Biochimica et Biophysica Acta (BBA) - Molecular Basis of Disease. 2014; 1842(3):446-462. https://www.sciencedirect.com/science/article/pii/S0925443913001798

- Shiho Fujisaka et al.Regulatory Mechanisms for Adipose Tissue M1 and M2 Macrophages in Diet-Induced Obese Mice. Diabetes. 2009; 58 (11): 2574–2582. https://doi.org/10.2337/db08-1475

- Guo, Y. et al. AGEs Induced Autophagy Impairs Cutaneous Wound Healing via Stimulating Macrophage Polarization to M1 in Diabetes. Sci Rep 6, 36416 (2016). https://doi.org/10.1038/srep36416

  1. Another issue concerns discussion. The authors point out that infiltrating macrophages have a pro-inflammatory phenotype. How did the authors show that these are precisely the infiltrating macrophages of adipose tissue? Is there a difference in the phenotype of resident macrophages and those migrating in adipose tissue? I think that on the basis of the available markers, the authors cannot reliably speak about infiltrating macrophages, just as they cannot reliably speak about a change in the M1 or M2 phenotype of adipose tissue macrophages. 
  • Gordon (2005) described that replenishment of tissue-resident macrophages at inflammatory state greatly depends on circulating precursors. Also, Lee (2014) announced that the immunological unusual feature of obesity is the increase in CD11c+ ATM numbers, and an in vivo labeling study figured out that the resident ATMs do not express CD11c but are newly recruited ATMs express CD11c. In our study, we confirmed that the CD11c+ ATM population was enhanced in HFD-fed groups compared to the NC group. Therefore, we have written the increase in infiltrating macrophages.

Related References

- Gordon, S., Taylor, P. Monocyte and macrophage heterogeneity. Nat Rev Immunol 5, 953–964 (2005). https://doi.org/10.1038/nri1733

- Byung-Cheol Lee, Jongsoon Lee. Cellular and molecular players in adipose tissue inflammation in the development of obesity-induced insulin resistance. Biochimica et Biophysica Acta (BBA) - Molecular Basis of Disease. 2014; 1842(3):446-462. https://www.sciencedirect.com/science/article/pii/S0925443913001798

We thank you again for your insightful comments on our paper.

                                                       Sincerely yours,

 Byung-Cheol Lee, M.D.& Ph.D.

Round 2

Reviewer 2 Report

Thanks for answers.

  1. If you think your statistical analysis method is suitable for your data, then leave it as it was in the original wording of this section. The Mann-Whitney test is not used for pairwise comparisons in post-hoc comparisons, so remove references to it. 
  2. However, as a matter of discussion, most of all, the statement that CD11c can be used as an M1 marker raises questions. It is worth noting that at present, many scientists have expressed an opinion against the mechanistic division of macrophages strictly into M1 and M2, in addition, the view on markers M1 and M2 has been revised. It seems to me that the most adequate approach is presented in the next article https://www.ncbi.nlm.nih.gov/pmc/articles/PMC4123412/ . Please note that the authors of this approach propose almost completely away from the use of surface markers, and even more so they mention that the use of CD11c is very problematic. I think all these contradictions should be mentioned in the Discussion section. 

I consider it necessary to add to the discussion section a modern view of the M1 / M2 paradigm in the nomenclature of macrophages, as well as the difficulties in using the CD11c marker as a marker of the M1 phenotype. 

Author Response

Thank you very much for considering our manuscript for publication. Your suggestions were very helpful to us, and we have incorporated those points into our revised manuscript.

The changes made to the manuscript are as follows:

[Reviewer 2]

  1. If you think your statistical analysis method is suitable for your data, then leave it as it was in the original wording of this section. The Mann-Whitney test is not used for pairwise comparisons in post-hoc comparisons, so remove references to it. 
  • We revised the part to the original text.

  1. However, as a matter of discussion, most of all, the statement that CD11c can be used as an M1 marker raises questions. It is worth noting that at present, many scientists have expressed an opinion against the mechanistic division of macrophages strictly into M1 and M2, in addition, the view on markers M1 and M2 has been revised. It seems to me that the most adequate approach is presented in the next article https://www.ncbi.nlm.nih.gov/pmc/articles/PMC4123412/ . Please note that the authors of this approach propose almost completely away from the use of surface markers, and even more so they mention that the use of CD11c is very problematic. I think all these contradictions should be mentioned in the Discussion section. 

I consider it necessary to add to the discussion section a modern view of the M1 / M2 paradigm in the nomenclature of macrophages, as well as the difficulties in using the CD11c marker as a marker of the M1 phenotype. 

  • As you commented, we added the issue about in-consensus on M1-M2 definition and inconsistent use of surface markers on macrophages at page 11, line 357, in discussion part.